# Do Disease and Pest Control Outsourcing Services Reduce Arable Land Abandonment? Evidence from China

**DOI:** 10.3390/ijerph191811398

**Published:** 2022-09-10

**Authors:** Xiaoheng Zhang, Guiquan Yan, Yucheng He, Hailong Yu

**Affiliations:** 1Circulation Industry Research Central of Chinese Capital, Beijing Technology and Business University, Beijing 100048, China; 2College of Economics and Management, Nanjing University of Aeronautics and Astronautics, Nanjing 211106, China; 3College of Economics and Management, Huazhong Agricultural University, Wuhan 430070, China

**Keywords:** arable land abandonment, DPCOS, CLDS, China

## Abstract

Arable land abandonment has been occurring in China in recent years. Although an emerging number of studies have investigated the impacts of urbanization and labor migration on arable land abandonment, little is known about what roles agricultural outsourcing services play in reducing arable land abandonment. Based on the data from the China Labor-force Dynamics Survey (CLDS) in both 2014 and 2016, this study employs a two-stage least squares method to address the potential endogeneity issue and sheds some light on the impact of agricultural outsourcing services for controlling disease and pests in arable land abandonment in China. The empirical results show that disease and pest control outsourcing services (DPCOS) significantly decrease the size of household-level arable land abandonment by 6.59% on average. More specifically, DPCOS mainly reduce the arable land abandonment in regions with the labor shortages, while this does not lead to a significant decrease in arable land abandonment in regions characterized by poor soil quality and steep slopes. Therefore, we may conclude that DPCOS could contribute to the reuse of farmlands suitable for cultivation and the exit of farmlands unsuitable for cultivation.

## 1. Introduction

The abandonment of arable land is a ubiquitous phenomenon in the world and happens in North America, Europe, Japan, and other developed and developing countries [1,2,3,4,5]. Many arable lands have been abandoned in China in recent years, especially in mountainous areas [5,6,7,8,9,10]. Li and Li pointed out that there are about 2 million hm^2^ of arable land falling out of production each year in China [11]. Li et al. documented that 14.32% of arable land located in China’s mountainous areas has been abandoned [12]. Land abandonment poses a great threat to China’s food security. China is a land-scarce country with only 8% of the world’s arable land but has to feed 20% of the world’s population [13,14,15,16]. In this context, China has witnessed an uptrend of marginal lands such as hills, uplands, and grasslands reclaimed for cultivation throughout the past decades [17,18], although these lands have been characterized by low productivity. China’s farmers are also encouraged to use more chemical fertilizers and pesticides to overcome land scarcity limitations [19]. China has achieved staple food self-sufficiency of 95% [20]. However, given rising income and rapid urbanization, it is increasingly urgent to consolidate the cornerstone of food security nationwide by reducing arable land abandonment and effectively using arable lands [13,21]. In addition, Benayas et al. pointed out that arable land abandonment may reduce landscape heterogeneity and water provision, increase fire frequency, magnify soil erosion and biodiversity loss, and give rise to loss of cultural and aesthetic value [22].

Disease and pest control outsourcing services (DPCOS) are an effective measure to reduce farmers’ arable land abandonment in China. The agricultural production process contains land preparation, seeding, disease and pest control, and harvesting. The disease and pest control process is labor and knowledge-intensive and is one of the most care-intensive processes for farmers. It is more difficult to substitute machinery for labor than the other processes, and it requires rich experience and professional knowledge [23,24]. With the emergence of DPCOS, farmers could buy the DPCOS rather than doing it by themselves to control disease and pests. DPCOS are usually provided by agricultural service organizations that employ many workers with specialized skills. Additionally, farmers need not rent out their arable lands by buying DPCOS. Based on DPCOS, farmers could contribute a few family laborers and their efforts to complete the production process effectively. Therefore, the purpose of this paper is to investigate the impact of DPCOS on arable land abandonment and to further explore the heterogeneous effects of DPCOS on arable land abandonment across households with different natural resource endowments. Based on the results, this paper raises some policy implications to reduce arable land abandonment. A few studies have investigated the reasons for arable land abandonment and studied the impacts of agricultural outsourcing services on agricultural production in China. Although the Chinese government has adopted a set of measures to prevent the loss of arable lands from non-agricultural competition, at least three reasons contribute to farmers’ arable land abandonment. First, it is worth noting that agriculture is becoming less attractive for many farmers [11,25,26,27,28]. With rapid urbanization and industrialization, a large proportion of farmers could take on off-farm jobs, and their agricultural income proportion decreased from 63.3% in 2000 to 36.7% in 2018. Agricultural operations are becoming sidelines for many farmers. Thus, farmers put less effort into their arable lands and even abandon their arable lands [29,30,31]. Second, nearly 291 million peasant workers in 2019, most of whom are young, migrated to work in cities while only children, women, and older people were left behind [5,6,8,11,32,33]. This led to low arable land use efficiency or even abandonment of arable land due to labor scarcity. Third, farmers abandon their arable lands rather than renting them out because Chinese arable land property rights are not well-defined [10,34,35]. It is worth noting that the Chinese government innovates land institutions by separating operational rights from the current contract rights [10,14,16]. This may eliminate farmers’ fear of losing arable land and promote arable land circulation.

In addition, agricultural outsourcing services have documented that it could increase technical efficiency [27,29,36], alleviate farmers’ labor shortage, decrease agricultural production costs, and improve agricultural benefits [37,38,39,40]. Agricultural outsourcing services include a set of service forms such as land preparation services, seeding services, harvesting services, DPCOS, agricultural technology extension services [37,39,41], and training and information services [38]. More specifically, Jiang et al., Yang et al., and Chen and Tang found that agricultural outsourcing services have a significant positive impact on farmers’ land renting decision-making [42,43,44]. Luo et al. further found that agricultural outsourcing services can significantly reduce farmers’ abandonment of arable land caused by land fragmentation [45].

This study contributes to the literature in three ways. First, given the threat of land abandonment to food security and the rapid development of agricultural outsourcing services in China, this paper takes China as a case to investigate the impact of agricultural outsourcing services on land abandonment. This paper focuses on the disease and pest control process, one of the most important processes of agricultural production and one being ignored by the current literature, and investigates the impacts of DPCOS on farmers’ arable land abandonment. This study also explores the heterogeneous impacts of DPCOS on arable land abandonment across rural households with differentiated labor endowments, soil, and agricultural terrain characteristics. The findings have important policy implications for the development of DPCOS. Second, the data used in this paper is from the China Labor-force Dynamics Survey (CLDS). Compared with some regional survey data, the CLDS is a nationally representative dataset and has rich information on both DPCOS and arable land abandonment [44,45]. Third, to the extent that DPCOS adoption is not randomly distributed, farmers may have some unobservable characteristics which are correlated with both arable land abandonment and DPCOS use decisions. Therefore, this paper adopts a two-stage least squares method (2SLS) to address the potential endogeneity issue. We use the development levels of DPCOS at the provincial level, city level, and county level as the instrumental variables because the higher the regional DPCOS level is, the higher the possibility for farmers to use DPCOS is.

We organize the remainder of this paper as follows. Section 2 provides an overview of DPCOS in China. Section 3 introduces the data and methodology. Section 4 presents the empirical results, and Section 5 concludes.

## 2. The Development of DPCOS in China

Agricultural outsourcing service is driven by the specialized division of labor in agricultural production and is an important innovation for China’s agriculture operational model. The notion of agricultural outsourcing services was firstly proposed by the No.1 Central Document in 1983. However, farmers’ demand for agricultural outsourcing services has remained low until recent years. At the end of 2018, 0.37 million organizations provided outsourcing services for 364 million mu (One hectare = 15 mu.) of sown areas [46,47]. Therefore, propelled by the mega-trend of young laborers migrating to urban areas for off-farm work, farmers have a high demand for agricultural outsourcing services. The agricultural outsourcing service organizations mainly help farmers to complete the production processes that are difficult or costly. To promote agricultural outsourcing services, the Chinese government subsidized 3 billion CNY and 4 billion CNY in 2017 and 2018, respectively [46].

DPCOS plays a pivotal role in agricultural outsourcing services [48]. The production processes of land preparation, seeding, and harvesting have a high level of mechanization due to the high substitution elasticity between machines and human labor in China [49,50]. However, disease and pest control need much effort to monitor and identify the disease and pests and need special skills and knowledge to control them [24,51,52]. It is a convenient measure to effectively control disease and pests by using DPCOS for farmers with labor scarcity and lack of knowledge. Therefore, DPCOS could alleviate arable land abandonment in China. MARA estimated that 39.2% of sown areas of rice, wheat, and corn were being conducted by DPCOS organizations to control disease and pests in 2017, with an increase of 1.4% over the prior year [53].

DPCOS is beneficial to the sustainability of China’s agriculture industry [23,54,55]. Ji documented that farmers’ production costs using agricultural outsourcing services could also decrease significantly from 20% to 40% [47]. On the one hand, the suppliers of DPCOS could achieve economies of scale by providing a large number of farmers with services for controlling disease and pests. On the other hand, DPCOS could lead to a 10% to 20%, even a 30% reduction, in the number of chemical pesticides per mu [56] by applying the agrichemicals in a timely and accurate manner with appropriate methods. DPCOS organizations aim at preventing the outbreak of disease and pests rather than how to treat them afterward based on specialized knowledge and the monitoring and early warning systems [48]. In addition, using DPCOS could promote the adoption of environmentally friendly techniques for disease and pest control [48,54] and reduce agrichemical exposure to farmers’ health [55].

## 3. Methodology and Data

This section may be divided into subheadings. It should provide a concise and precise description of the experimental results, their interpretation, as well as the experimental conclusions that can be drawn.

### 3.1. Methodology

To investigate the impact of DPCOS on arable land abandonment, a baseline model takes the form:(1)$$y_{\mathrm{it}}=\alpha_{1}{\mathrm{DPCOS}}_{\mathrm{it}}+\alpha_{2}Z_{1}+e$$
where $y_{\mathrm{it}}$ denotes the ratio of arable land abandonment, ${\mathrm{DPCOS}}_{\mathrm{it}}$ is the key explanatory variable and equals 1 if farmer i using DPCOS in year t, and 0 otherwise. $Z_{1}$ is a matrix of control variables, including the characteristics of household head, household, and village. $\alpha_{1}$ and $\alpha_{2}$ are the coefficients to be estimated, and e is the error term. We adopt a Tobit estimator since the ratio of arable land abandonment is between 0 and 100, and we also regress Equation (1) by using the OLS linear probability model (Though linear probability models run the danger of allowing probabilities to be higher than 100 (or less than 0), these do not substantially affect the main results. Therefore, the OLS linear probability model is (approximatively) reasonable). This study includes city dummy variables and a time dummy variable in the model.

The baseline model may suffer from endogeneity problems due to the following reasons: First, Equation (1) may omit some unobserved factors that are correlated with both DPCOS adoption decision and arable land abandonment decision, such as innate abilities (hedonic personality) and motivation to abandon the arable land. Second, DPCOS adoption decision and arable land abandonment decision may have reverse causation. Using DPCOS may reduce farmers’ arable land abandonment, but the high level of arable land abandonment may induce the development of DPCOS.

A two-stage least squares method is a methodology to address the potential endogeneity issue. Inspiring from the rationale of peer effects and externalities in technological adoption [45,57], this study sets the provincial level of DPCOS as the instrumental variable (IV) for DPCOS. It is measured by the average value of DPCOS for the counties except for the sample county in the same province. This IV is directly related to DPCOS but does not relate to the households’ decision of arable land abandonment. Although the cross-regional agricultural mechanization services experienced rapid growth in China [58], DPCOS mainly serves local farmers.

### 3.2. Data

The data used in this study are from the China Labor-force Dynamics Survey (CLDS) in 2014 and 2016 (The CLDS data for 2020 has not been released yet. Although we have learned that CLDS released the data in 2018, we can only get a small part of the information from 2018. The new samples in 2018 cannot be combined with the original follow-up samples from 2014 and 2016 to form balanced panel data, which will have a greater impact on the undertaken research. Therefore, we only used the original follow-up samples from 2014 and 2016), collected by the Center for Social Science Survey of Sun Yat-Sen University. CLDS was launched in 2011 and is conducted every 2 years in 29 Chinese provinces in China, except for Hong Kong, Macao, Taiwan, Tibet, and Hainan. The stratified random sampling procedure is adopted to determine the sample of the CLDS to ensure the representativeness of the sample. The dataset contains information about individuals, households, and villages. More specifically, the detailed information includes the characteristics of individuals, such as education, work conditions, migration, health, social participation, economic activities, and other domains (More information is available at http://css.sysu.edu.cn, accessed on 23 August 2019).

Table 1 lists the variable definitions. The dependent variable is measured by the ratio of the abandoned area to the total area at the household level. Given the availability of household-level information on DPCOS, this study examines whether the villages have DPCOS as the proxy variable for farmers. Based on the literature, the control variables include household head, household, and rural village characteristics.

Table 2 presents summary statistics. The results show that the ratio of arable land abandonment in our sample is 7.35%. It indicates that 135 million mu of arable lands might be abandoned in China. Therefore, arable land abandonment poses a severe threat to Chinese food security. In addition, we find that 37.6% of villages in our sample have adopted DPCOS.

## 4. Empirical Results and Discussion

### 4.1. Baseline Results

The estimated results of the DPCOS’s effect on arable land abandonment are reported in Table 3. First, we used the OLS and Tobit model to estimate Equation (1). The results show that DPCOS has no significant effect on arable land abandonment regardless of whether the control variables are included or not. A possible explanation is that the variable of DPCOS may suffer from an endogeneity issue.

The under-identification test and weak identification are reported in the fourth and fifth rows in Table 4. The Kleibergen–Paap rk LM statistics in the first stage of the 2SLS estimates are significant at the 1% significance level, implying that the instrumental variables can explain the variation of endogenous variables. Moreover, the Montiel–Pflueger robust weak instrument statistic in the first stage of the 2SLS estimates also rejects the hypothesis that bias is more than 5% of “worst-case” (completely uninformative instruments) bias at the 1% significance level, indicating that the provincial level of DPCOS is not a weak instrument for the village-level DPCOS.

Columns (5) and (6) in Table 4 report the results of the 2SLS fixed-effects specification. The results of Wooldridge’s endogeneity test indicate that DPCOS is not exogenous. The estimated results in Column (6) indicate that compared with the villages that did not use any DPCOS, the villages adopting DPCOS have a statistically significant decrease in the arable land abandonment ratio by an average of 6.59%. In Column (5), we also find that the coefficient of DPCOS variable without control variables is negative and statistically significant at the 1% significance level, indicating that the adoption of DPCOS generated about a 5.52% decrease in arable land abandonment. These results are consistent with the findings by Luo et al. [45]. In addition, we find that the extent of the impact of DPCOS on land abandonment will increase as the control variables are added.

### 4.2. Robustness Test

In this subsection, we check the robustness of our results in several ways. More specifically, we change IV by using the other agricultural outsourcing services and we also change the dependent variable. 

#### 4.2.1. Changing the IV

To test the robustness of the results, we replace IV with the average value of village DPCOS, excluding the sample village within the same municipal level or county level. The results are reported in Table 5. Columns (3) and (4) report the estimated results by using the IV corresponding to city-level DPCOS (Panel A) and county-level DPCOS (Panel B), respectively. The results are similar to the results in Table 4. Table 5 also presents the results of the under-identification test and weak identification test for the instrumental variables. 

#### 4.2.2. Including the Other Agricultural Outsourcing Services

Since the other agricultural outsourcing services may also affect farmers’ land use behavior, the baseline results may overestimate the impact of DPCOS on arable land abandonment. In addition, farmers may book agricultural outsourcing services, including both DPCOS and the other services, which would make it difficult to separate the impacts of DPCOS from the total effects. Therefore, we try our best to precisely evaluate the impacts by including the other agricultural outsourcing services into our model, such as irrigation and drainage services, mechanized planting services, agricultural material supply services, planting programming services, employment guidance services, and agricultural technology training services. We find that the results reported in Table 6 remain robust.

#### 4.2.3. Using Rural-Village-Level Rations of Arable Land Abandonment as Dependent Variables

In this section, we replace rations of arable land abandonment on the household level with that on the village level, and the results are reported in Table 7. We find that the village-level DPCOS could also significantly reduce arable land abandonment, consistent with the results mentioned above.

### 4.3. Heterogeneity Test

The DPCOS may have differentiated effects on arable land abandonment in regions with different labor resources and natural resource endowments. This subsection investigates the heterogeneity effects of DPCOS on arable land abandonment.

#### 4.3.1. Heterogeneous Effects with Respect to Labor Resource Endowment

Driven by rapid urbanization, many youngsters migrate to cities and engage in off-farm work [42]. In this context, agricultural outsourcing services have emerged to help farmers operate their farmlands [24,29,44]. Thus, would the DPCOS only affect the households or regions with labor shortages? We present the results in Table 8.

We divide the sample into five groups according to the quintiles of the non-agricultural employment ratio (It is measured by the proportion of family members engaged in non-agricultural production for more than 3 months per year). Although we find that DPCOS has statistically significant impacts on arable land abandonment for each group except for the middle-low group, DPCOS will reduce arable land abandonment for middle, middle-high, and high groups by 3.70%, 7.50%, and 21.59%, respectively. It is consistent with our expectation that DPCOS reduce arable land abandonment by alleviating the farmers’ labor shortage.

To further evaluate the effect of DPCOS on arable land abandonment, we conducted the regression on the village level by separating the villages into five groups according to the ratio of the seasonal non-agricultural employment on the village level. The results are presented in Panel B in Table 8. As shown in Column (2), DPCOS could only reduce arable land abandonment in villages with middle-high and high ratios of seasonal non-agricultural employment but have no impact among villages with low, middle-low, and middle seasonal non-agricultural employment. 

#### 4.3.2. Heterogeneous Effects with Respect to Natural Resource Endowment

The conditions correlated with natural resource endowments, such as poor soil quality and vulnerable marginal agricultural terrain, may be uncertain in the association of DPCOS with arable land abandonment. This study further sheds some light on whether the impacts of DPCOS on the households’ decision to abandon arable land vary over various cohorts partitioned by natural characteristics.

Soil degradation, following the trajectories of erosion, sedimentation, and salinization, emerges as both a consequence of inappropriate agricultural practices and a problem linked to arable land abandonment [17,22]. In light of this, based on implementing the Soil Modification Programme (SMP), we compare the villages with and without soil recovery in Panel A of Table 9 (The thought behind this grouping is that the given villages, likely confronted with unsuitable soil nutrients and severe soil degradation from short- and long-term land mismanagement may use soil modification techniques to improve soil fertility). We find that DPCOS would reduce arable land abandonment in villages that did not implement the Soil Modification Programme (SMP), and it is not the case for villages that implemented SMP. One possible explanation is that villages that implement SMP may suffer from severe soil degradation and their lands are unsuitable for crop cultivation. Therefore, DPCOS does not affect arable land abandonment triggered by soil degradation.

Increased intensification in arable land and agricultural operation occurs alongside land abandonment and afforestation, which can result in land abandonment [22]. To shed more light on the heterogeneous responses of arable land abandonment to DPCOS across the cohorts with the different agricultural intensification, we explore whether large and significant mitigative effects for the village with intensive utilization of arable land. In this connection, we apply the Grain-for-Green Programme (GGP) to compare the differential effects of DPCOS by focusing on heterogeneity between the villages with and without GGP (GGP was initiated in Sichuan, Shaanxi, and Gansu in 1999 and roundly launched in 2002. To bring water loss and soil erosion under control by conversing inefficient land use patterns of marginal arable land with sustainable land use patterns [59], the Chinese government impels this nationwide initiative of payments for ecosystem services. Soil erosion is bound up with land use patterns and agricultural landscapes; thus, the vulnerable margin of arable land on steep slopes is the bull’s eye of GGP [59]). Typically, after being included in the scope of GGP, the villages shift toward the appropriate scale operation, while the land use pattern converts to intensification [59,60]. 

Panel B presents estimates for the villages with and without GGP. Column (2) shows that the adoption of DPCOS statistically significantly decreases the ratio of arable land abandonment by 5.68% in the villages with GGP, whereas DPCOS has no effects in the villages without GGP. It suggests that DPCOS may be of no help to the arable land abandonment caused by extensive utilization patterns in marginal land and fragile agricultural landscapes but to those by agricultural intensification, which is in line with the studies on the associations between agricultural outsourcing services and arable land renting [42,43,45]. The Closing-Hillsides-for-Afforestation Programme (CHAP) shares a high degree of homology to GGP. Applying the CHAP in Panel C, we also obtain similar results with Panel B.

Elevation, geological substrate, and slope are nonnegligible ecological drivers of arable land abandonment [22]. Compared with hilly and plain terrain, mountainous terrain is rugged and rough and hinders DPCOS operations. In addition, such mountainous terrain is not conducive to replacing labor with a machine that is the basis of DPCOS. The opportunity cost of on-farm laborers and the price of DPCOS are also susceptible to the agricultural terrain. Panel D presents the results for the given villages’ characteristics of different terrain features. We find that the positive impacts of DPCOSs on reducing arable land abandonment may be offset by the vulnerable marginal agricultural terrain, which is not suitable for crop cultivation. In general, we may conclude that DPCOS could contribute to the reuse of lands suitable for crop cultivation and the exit of lands unsuitable for crop cultivation. Therefore, DPCOS also contribute to the formation of multifunctional landscapes discussed by Sturck and Verburg and De Groot et al. [61,62].

## 5. Conclusions

Efficiently using arable land plays a crucial role in ensuring China’s food security. However, many arable lands have been abandoned in China, especially in the mountainous areas. Agricultural outsourcing services have been prioritized in the developmental trajectories of modern agriculture in China. Although an emerging number of studies have investigated the impacts of urbanization and labor migration on arable land abandonment, little is known about what roles agricultural outsourcing services play in reducing arable land abandonment. As one of the most care-intensive processes for farmers, the disease and pest control process need much effort to monitor and identify the disease and pests and need special skills and knowledge to control them. It is a convenient measure to effectively control disease and pests by using DPCOS for farmers with labor scarcity and lack of knowledge. This study aimed at investigating the impacts of DPCOS on arable land abandonment in China.

The key findings show that DPCOS has a negative and statistically significant effect on arable land abandonment by an average of 6.59%, with the endogeneity problem a consideration. Moreover, we also conduct the heterogeneity test on whether the effects of DPCOS on arable land abandonment vary over the characteristics of labor endowment and natural endowment. The results show that DPCOS mainly reduces the arable land abandonment in regions with labor shortages, while it does not lead to a significant decrease in arable land abandonment in regions characterized by poor soil quality and steep slopes. Therefore, promoting the development of DPCOS is not only important for Chinese agriculture but also provides significant policy implications for policymakers in developing countries.

Based on the empirical results, this study provides some valuable implications for local governments. First, promoting the development of disease and pest control outsourcing services could reduce arable land abandonment which would threaten food security in China. Second, the focus of the policy is to encourage areas suitable for grain cultivation to reuse abandoned land through promoting the development of disease and pest control outsourcing services while encouraging areas unsuitable for cultivation to exit agricultural operations.

This study has limitations. There are no household-level data on soil quality and topography in the CLDS dataset, and we have to use village-level data as a proxy. Therefore, all households within a village face the same soil quality and topography. We would further re-study this topic if we can match CLDS dataset with Harmonized World Soil Database using the longitude and latitude of each household.

## Figures and Tables

**Table 1 ijerph-19-11398-t001:** Definitions of variables.

Variable	Implication and Value
The ratio of arable land abandonment	The rate of the abandoned area to the total area at the household level
Disease and pest control outsourcing service (DPCOS)	1 = yes, 0 = no
Gender of household head (HH)	1 = male, 0 = female
Education level of HH	Years
Hukou of HH	1 = agricultural hukou or urban-rural unified hukou ^1^ transformed from agricultural hukou, 0 = non-agricultural hukou or urban-rural unified hukou transformed from non-agricultural hukou
Health of HH	1 = very good health, 2 = good health, 3 = general, 4 = poor health, 5 = very poor health
Age of HH	Age of household head
Tractor	Whether the household owns tractors? (1 = yes, 0 = no)
Large agricultural machinery	Whether the household owns large agricultural machinery (i.e., harvester, transplanter, drill, large combine harvester) (1 = yes, 0 = no)
Government subsidy	Whether the household obtains subsidies (1 = yes, 0 = no)
Specialized household ^2^	Whether the household is specialized household? (1 = yes, 0 = no)
income	Average annual income per household member
Farmland registration	Whether the household owns the certificate of Right to Land Contractual Management in Rural Area (1 = yes, 0 = no)
Average farmland (mu)	Area of farmland per household member
Suburb	Whether the village is located in a suburb (1 = yes, 0 = no)
Non-agricultural economy	Whether the village has a non-agricultural economy (1 = yes, 0 = no)
Natural disaster	Has the village ever suffered a serious natural disaster (1 = yes, 0 = no)
Distance to the county center (km)	The logarithm of the distance between the nearest county center or district government and rural village
Terrain	1 = plain terrain, 2 = hilly terrain, 3 = mountainous terrain

Note: ^1^ Hukou refers to the household registration system in China. There is a huge gap between the rural and urban areas in terms of public services, and rural residents can only enjoy urban public services if they get an urban hukou. To control the migration of rural residents to an urban area, China established the hukou system. China has reformed its hukou system since 2014, and a rural-urban unified hukou system has been implemented in some pilot cities. ^2^ Specialized households in agricultural production: A group of farmers in rural areas of China specialized or mainly engaged in certain production activities (i.e., pisciculture, flower-cultivating, apiculture, pig-breeding, sericulture). Unlike ordinary farmers, specialized households are oriented by sales, and their production scale is far larger than the average level of ordinary farmers. Mu is a unit of measurement of land area in China, and fifteen mu is equal to 1 hectare.

**Table 2 ijerph-19-11398-t002:** Summary of variables.

	Obs.	Mean	Sta. Dev.	Min	Max
The ration of arable land abandonment	15,638	7.354	23.536	0	100
DPCOS	15,694	0.376	0.484	0	1
Gender of HH	22,223	0.882	0.323	0	1
Education level of HH	22,009	11.325	5.534	0	22
Hukou of HH	22,212	0.959	0.198	0	1
Health of HH	22,183	2.469	1.050	1	5
Age of HH	21,928	53.576	13.07	3	108
Large agricultural machinery	22,268	0.021	0.143	0	1
Tractor	22,272	0.132	0.338	0	1
Government subsidy	10,070	0.599	0.490	0	1
Specialized household	10,072	0.079	0.270	0	1
Average annual income per household member	20,496	8.361	1.192	−0.102	13.364
Farmland registration	14,361	0.488	0.500	0	1
Area of farmland per household member	21,384	1.398	5.327	0	511
Suburb	22,354	0.082	0.275	0	1
Non-agricultural economy	21,844	0.321	0.467	0	1
Natural disaster	21,879	0.466	0.499	0	1
Distance to the county center	21,701	25.625	21.805	0	120
Terrain	20,193	1.811	0.836	1	3

**Table 3 ijerph-19-11398-t003:** Baseline results: The impact of DPCOS on arable land abandonment.

	OLS (Y = Arable Land Abandonment)		Tobit (Y = Arable Land Abandonment)	
	(1)	(2)	(3)	(4)
DPCOS	−0.54	−0.07	−0.54	−0.07
	(−1.05)	(−0.14)	(−1.06)	(−0.14)
Observations	14,871	6860	14,871	6860
Controls	No	Yes	No	Yes
City dummies	Yes	Yes	Yes	Yes
Year dummies	Yes	Yes	Yes	Yes

Note: Control variables include household head characteristics (gender, education level, hukou, health, age), household characteristics (tractor, large agricultural machinery, government subsidy, specialized household, income, farmland registration, average farmland), and village-level characteristics (non-agricultural economy, natural disaster, terrain). The numbers in brackets are robust standard errors.

**Table 4 ijerph-19-11398-t004:** Baseline results: The impact of DPCOS on arable land abandonment by using 2SLS.

	First Stage (Y = DPCOS)		Reduced-Form Estimates (Y = Arable Land Abandonment)		The Second Stage (Y = Arable Land Abandonment)	
	(1)	(2)	(3)	(4)	(5)	(6)
DPCOS					−5.52 ***	−6.59 ***
					(−6.34)	(−4.48)
Provincial-level DPCOS	−4.21 ***	−4.71 ***	22.49 ***	31.04 ***		
	(−75.02)	(−24.55)	(6.28)	(4.43)		
K-P	3666.55 ***	653.58 ***				
M-P	4883.74 ^a^	707.779 ^a^				
Wooldridge endogeneity test					29.64 ***	19.33 ***
Observations	14,630	6738	15,154	6738	14,630	6738
Controls	No	Yes	No	Yes	No	Yes
City dummies	Yes	Yes	Yes	Yes	Yes	Yes
Year dummies	Yes	Yes	Yes	Yes	Yes	Yes

Notes: Control variables include household head characteristics (gender, education level, hukou, health, age), household characteristics (tractor, large agricultural machinery, government subsidy, specialized household, income, farmland registration, average farmland), and village-level characteristics (non-agricultural economy, natural disaster, terrain). *** represent statistical significance level of 1%. The numbers in brackets are robust standard errors. K-P refers to the Kleibergen–Paap rk LM statistic in the first stage of the 2SLS estimates. M-P refers to the Montiel–Pflueger robust weak instrument statistic in the first stage of the 2SLS estimates. Wooldridge endogeneity test refers to the robust score test statistic. ^a^ The hypothesis that bias is more than 5% of ‘worst-case’ (completely uninformative instruments) bias is rejected at the 1% significance level.

**Table 5 ijerph-19-11398-t005:** Estimate results from augmented IVs.

	First Stage (Y = DPCOS)		Second Stage (Y = Arable Land Abandonment)	
	(1)	(2)	(3)	(4)
Panel A. City-Level DPCOS as IV for Rural Village DCPCOS				
City-level DPCOS	0.12 ***	0.10 ***		
	(41.56)	(18.42)		
DPCOS			−8.26 ***	−6.04 ***
			(−6.01)	(−3.06)
Observations	13,696	6235	13,696	6235
K-P	1120.66 ***	263.18 ***		
M-P	1728.00 ^a^	339.90 ^a^		
Panel B. County-level DPCOS as IV for Rural Village DCPCOS				
County-level DPCOS	0.11 ***	0.08 ***		
	(33.23)	(13.84)		
DPCOS			−9.78 ***	−10.71 ***
			(−6.16)	(−3.91)
Observations	14,253	6497	14,253	6497
K-P	814.62100 ***	166.89 ***		
M-P	1104.72 ^a^	191.87 ^a^		
Observations	9648	4279	9648	4279
Controls	No	Yes	No	Yes
City dummies	Yes	Yes	Yes	Yes
Year dummies	Yes	Yes	Yes	Yes

Note: Control variables include household head characteristics (gender, education level, hukou, health, age), household characteristics (tractor, large agricultural machinery, government subsidy, specialized household, income, farmland registration, average farmland), and village-level characteristics (non-agricultural economy, natural disaster, terrain). *** represent statistical significance level of 1%. The numbers in brackets are robust standard errors. K-P refers to the Kleibergen–Paap rk LM statistic in the first stage of the 2SLS estimates. M-P refers to the Montiel–Pflueger robust weak instrument statistic in the first stage of the 2SLS estimates. ^a^ The hypothesis that bias is more than 5% of ‘worst-case’ (completely uninformative instruments) bias is rejected at the 1% significance level.

**Table 6 ijerph-19-11398-t006:** Inclusion of other AOSs.

	(1)	(2)	(3)	(4)	(5)	(6)
Panel A. Irrigation and Drainage Services						
DPCOS	−7.53 ***					
	(−4.21)					
Panel B. Mechanized Planting Services						
DPCOS		−7.21 ***				
		(−4.45)				
Panel C. Agricultural Material Supply Services						
DPCOS			−6.76 ***			
			(−4.55)			
Panel D. Planting Programming Services						
DPCOS				−10.11 ***		
				(−3.52)		
Panel E. Employment Guidance Services						
DPCOS					−6.60 ***	
					(−4.48)	
Panel F. Agricultural Technology Training Services						
DPCOS						−6.55 ***
						(−4.48)
Observations	6738	6738	6738	4871	6738	6738
Controls	Yes	Yes	Yes	Yes	Yes	Yes
City dummies	Yes	Yes	Yes	Yes	Yes	Yes
Year dummies	Yes	Yes	Yes	Yes	Yes	Yes

Note: Control variables include household head characteristics (gender, education level, hukou, health, age), household characteristics (tractor, large agricultural machinery, government subsidy, specialized household, income, farmland registration, average farmland), and village-level characteristics (non-agricultural economy, natural disaster, terrain). *** represent statistical significance level of 1%. The numbers in brackets are robust standard errors.

**Table 7 ijerph-19-11398-t007:** Replacing core outcome variable.

	First Stage (Y = DPCOS)		Second Stage (Y= Village-Level Arable Land Abandonment)	
	(1)	(2)	(3)	(4)
Provincial-level DPCOS	−4.35 ***	−4.36 ***		
	(−36.99)	(−34.01)		
DPCOS			−3.44 ***	−2.26 ***
			(−4.84)	(−4.42)
Observations	14,847	12,635	14,847	12,635
K-P	555.70 ***	531.36 ***		
M-P ^a^	1249.45	1180.05		
Controls	No	Yes	No	Yes
City dummies	Yes	Yes	Yes	Yes
Year dummies	Yes	Yes	Yes	Yes

Note: Control variables include village-level characteristics (non-agricultural economy, natural disaster, terrain). *** represent statistical significance level of 1%. The numbers in brackets are robust standard errors. K-P refers to the Kleibergen–Paap rk LM statistic in the first stage of the 2SLS estimates. ^a^ M-P refers to the Montiel–Pflueger robust weak instrument statistic in the first stage of the 2SLS estimates and rejects hypothesis that bias is more than 5% of ‘worst-case’ (completely uninformative instruments) bias at the 1% significance level.

**Table 8 ijerph-19-11398-t008:** The heterogeneity of labor resource endowment.

	2SLS (Y = Arable Land Abandonment)	
	(1)	(2)
Panel A. The Ratio of the Non-Agricultural Employment of Rural Households		
Low group	−3.34	−4.41 **
	(−1.40)	(−1.97)
Middle–low group	−0.92	−2.56
	(−0.27)	(−0.99)
Middle group	−2.43	−3.70 *
	(−1.08)	(−1.91)
Middle–high group	−8.41 **	−7.50 ***
	(−2.42)	(−2.62)
High group	−15.07 ***	−21.59 *
	(−4.28)	(−1.75)
Panel B. The Village-Level Ratio of the Seasonal Non-Agricultural Employment		
Low group	−6.86 **	−3.14
	(−2.50)	(−0.42)
Middle–low group	−2.11	−2.84
	(−0.32)	(−0.95)
Middle group	−6.64	2.12
	(−1.06)	(−0.79)
Middle–high group	−11.51 ***	−7.43 **
	(−3.35)	(−2.46)
High group	−10.09 ***	−11.56 ***
	(−2.85)	(−3.83)

Note: Control variables include household head characteristics (gender, education level, hukou, health, age), village-level characteristics (tractor, large agricultural machinery, government subsidy, specialized household, income, farmland registration, average farmland), and rural village characteristics (non-agricultural economy, natural disaster, terrain). *, **, and *** represent statistical significance at 10%, 5%, and 1%, respectively. Numbers in brackets are robust standard error.

**Table 9 ijerph-19-11398-t009:** The heterogeneity of natural resource endowment.

	2SLS (Y = Arable Land Abandonment)	
	(1)	(2)
Panel A. Soil Modification Programme (SMP) in the Given Villages		
Yes	1.37	4.10
	(0.08)	(0.15)
No	−5.88 ***	−2.71 ***
	(−5.50)	(−3.02)
Panel B. Grain-for-Green Programme (GGP)in the Given Villages		
Yes	−10.97 ***	−5.68 ***
	(−5.23)	(−3.82)
No	−3.02 ***	−1.35
	(−2.72)	(−1.10)
Panel C. Closing-Hillsides-for-Afforestation Programme (CHAP) in the Given Villages		
Yes	−6.85 ***	−6.40 ***
	(−2.92)	(−3.13)
No	−3.86 ***	−1.02
	(−3.09)	(−0.97)
Panel D. Village Terrain Features		
Plain terrain	−7.86 *	−8.92 ***
	(−1.92)	(−2.97)
Hilly terrain	−17.15 ***	−17.87 **
	(−3.86)	(−2.12)
Mountainous terrain	0.20	−1.44
	(0.09)	(−0.85)
Controls	No	Yes
City dummies	Yes	Yes
Year dummies	Yes	Yes

Note: Control variables include household head characteristics (gender, education level, hukou, health, age), village-level characteristics (tractor, large agricultural machinery, government subsidy, specialized household, income, farmland registration, average farmland), and rural village characteristics (non-agricultural economy, natural disaster, terrain). *, **, and *** represent statistical significance level of 10%, 5%, and 1%, respectively. Numbers in brackets are robust standard error. The middle–low group lacks enough observations.

## Data Availability

The data presented in this study are available upon request from the responding author.

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
