# Peer review of "Do Disease and Pest Control Outsourcing Services Reduce Arable Land Abandonment? Evidence from China"

_ijerph, 2022, doi:10.3390/ijerph191811398_

Round 1

Reviewer 1 Report

1. Better explanation of "Hukou" in Table 1.. Maybe you could mentioned it in Background.

2. Line 238 - why is Lou et al. linked.. underlined?

3. Give some Policy measures proposal in Conclusion

Author Response

Response to Reviewer 1 Comments

Point 1. Better explanation of "Hukou" in Table 1.. Maybe you could mentioned it in Background.

Response 1: Thank you very much for this comment. The purpose of this paper is to investigate the impact of disease and pest control outsourcing services on arable land abandonment in China, and the Hukou of the household head is a control variable. Therefore, we have added a note under Table 1 to provide more details on Hukou.  The added information is as follows: Hukou refers to the household registration system in China. There is a huge gap between the rural and urban areas in terms of public services, and rural residents can only enjoy urban public services if they get an urban hukou. China established the Hukou system to control the migration of rural residents to urban areas. China has reformed its hukou system since 2014, and a rural-urban unified hukou system has been implemented in some pilot cities.

Point 2. Line 238 - why is Lou et al. linked.. underlined?

Response 2: Thank you very much for pointing out this mistake. We have dropped the underline in Line 238 and have also dropped other underlines in the revised manuscript.

Point 3. Give some Policy measures proposal in Conclusion.

Response 3: Thank you very much for your suggestion. We have added some policy implications based on the empirical results in the Conclusion section. The added content is as follows: Based on the empirical results, this study provides some valuable implications for local governments. First, promoting the development of disease and pest control outsourcing services could reduce arable land abandonment which would threaten food security in China. Second, the focus of the policy is to encourage areas suitable for grain cultivation to reuse abandoned land through promoting the development of disease and pest control outsourcing services, while encouraging areas unsuitable for cultivation to exit agricultural operations.

Reviewer 2 Report

The article is very interesting and the statistics confirm the authors' conclusions. In the discussion, however, I miss the authors' statement on the relationship of land abandonment to subsidy schemes. Does the reduction of the subsidy, or the drop in the prices of crops, or the contamination of the soil, for example, have an effect on abandonment?

At the same time, I would be interested in the question of the relationship between the construction of new industrial plants in the region and the abandonment of land (people leaving for industry).

Based on the authors' analysis, I recommend adding to the discussion publications focused on integrated landscape management, for example: doi: 10.1007/s10980-016-0459-6, doi: 10.1016/j.ecocom.2009.10.006, doi: 10.3390/su541745

Author Response

Response to Reviewer 2 Comments

Point 1.The article is very interesting and the statistics confirm the authors' conclusions. In the discussion, however, I miss the authors' statement on the relationship of land abandonment to subsidy schemes. Does the reduction of the subsidy, or the drop in the prices of crops, or the contamination of the soil, for example, have an effect on abandonment?

Response 1: Thank you very much for your suggestion. In order to save space, we did not report the coefficients of the control variables. The results show that government subsidy has no effect on land abandonment. In addition, the results reported in Table 9 show that disease and pest control outsourcing service (DPCOS) would reduce arable land abandonment in villages that did not implement the Soil Modification Programme (SMP), and it is not the case for villages that implemented SMP.  One possible explanation is that villages that implement SMP may suffer from severe soil degradation and their lands are unsuitable for crop cultivation. Therefore, DPCOS has no effect on arable land abandonment in these villages.

Point 2. At the same time, I would be interested in the question of the relationship between the construction of new industrial plants in the region and the abandonment of land (people leaving for industry).

Response 2: Thank you very much for providing us with a new and interesting research topic. In my opinion, the construction of new industrial plants will result in the abandonment of land due to the higher wage of the industry. In fact, one of the reasons for land abandonment in rural China is the lower income from the agriculture sector compared to the non-agriculture sector. We controlled the non-agricultural economy on the village level in our empirical model by including a variable that whether the village has a non-agricultural economy. The empirical results show that a non-agricultural economy in the village has no effect on arable land abandonment. One possible explanation is that farmers are able to balance agricultural production and off-farm work if their off-farm works are in their villages. We will study the relationship between the construction of new industrial plants in the region and the abandonment of land in future studies.

Point 3. Based on the authors' analysis, I recommend adding to the discussion publications focused on integrated landscape management, for example: doi: 10.1007/s10980-016-0459-6, doi: 10.1016/j.ecocom.2009.10.006, doi: 10.3390/su541745

Response 3: Thank you very much for this comment. We have cited the former two recommended pieces of literature by Sturck and Verburg (2016) and Groot et al. (2010) in the revised manuscript. We don’t find the third literature using the doi: 10.3390/su541745.

Reviewer 3 Report

The weakness of the work is the poor setting of the discussed research issue in the literature, and the related lack of justification for what research gap the authors want to fill when conducting their research. There is also no complete discussion of the obtained results with the results of other authors. Doubts are raised in the order of the contents (especially in the introduction), moreover, in the article many issues and research steps taken are incompletely described, which raises a number of doubts and contributes to the low evaluation of this manuscript. The detailed comments are as follows:

The abstract lacks basic information about the methods used or the time scope of the research.

I suggest you think about the keywords, do they really reflect the essence of the work (e.g. Terrain ???)

In my opinion, the introduction requires ordering the content and a thorough change in the layout. In the introduction, one should refer to the discussed research problem and how it is recognized in the literature, not limiting ourselves to research / results concerning only China. Does the problem only concern China, if only the authors drew attention to it? There is also no indication of what gap in the research the authors want to fill by choosing China as the subject of research, which may raise doubts given the multiplicity of research presented by the authors in the introduction. After reading the introduction, a general question arises what is really important for the Authors: land abandonment or Disease and pest control outsourcing service? In the introduction, conclusions appear (lines 107-113), without any indication on what basis they are formulated.

The purpose of the work is blurred.

The title of part 2 is very laconic; taking into account the content of this part, the question arises what is its role?

There is no characterization of the study area (see comment below on ‘Terrain’) in the context of the research, which should be clearly visible in the text.

The use of data from 2014 and 2016 raises doubts. What justifies the use of historical data in the undertaken research?

Contents of table 1: 'Hukou ?; mu'? Even though there is a reference (Hukou) under the table, it needs to be improved and better related to the contents of the table.

In table 1 –‘Terrain’ - what the Authors mean; how the split was made. How does this relate to the 29 provinces that were available in the database used?

There is no discussion of the results.

In the manuscript, I miss an indication of the limitations the Authors see in their research.

The work requires intensive improvement of the text editing.

Author Response

Response to Reviewer 3 Comments

Point 1: The abstract lacks basic information about the methods used or the time scope of the research.

Response 1: We are very sorry for not presenting the basic information about the methods and the time scope of the research in our manuscript clearly. This paper employed a two-stage least-squares instrumental variable method to address the potential endogeneity issue and used the data from China Labor-force Dynamics Survey (CLDS) in both 2014 and 2016. We have added the basic information in the abstract section of the revised manuscript.

Point 2: I suggest you think about the keywords, do they really reflect the essence of the work (e.g. Terrain ???)

Response 2: Thank you very much for this comment. The terrain really does not reflect the essence of our work. Following your suggestion, we have deleted terrain, labor shortage, and soil quality, and have added CLDS and China as Keywords. CLDS is the dataset used in this paper.

Point 3: â‘ In my opinion, the introduction requires ordering the content and a thorough change in the layout. â‘¡In the introduction, one should refer to the discussed research problem and how it is recognized in the literature, not limiting ourselves to research / results concerning only China. Does the problem only concern China, if only the authors drew attention to it? There is also no indication of what gap in the research the authors want to fill by choosing China as the subject of research, which may raise doubts given the multiplicity of research presented by the authors in the introduction. â‘¢After reading the introduction, a general question arises what is really important for the Authors: land abandonment or Disease and pest control outsourcing service? â‘£In the introduction, conclusions appear (lines 107-113), without any indication on what basis they are formulated.

Response 3: Thank you very much for this constructive suggestion. â‘ We have reorganized the Introduction section. We describe the status of arable land abandonment and its impacts in China in the first paragraph. The second paragraph introduces the potential influence mechanism of disease and pest control outsourcing service (DPCOS) on arable land abandonment and raises our research question. The third and fourth paragraphs review the literature. The fifth paragraph shows the contributions. The last paragraph presents the structure of the paper.

â‘¡ The abandonment of arable land is a ubiquitous phenomenon in the world, and happens in North America, Europe, Japan, and other developed and developing countries (MacDonald et al., 2000; Parody et al., 2001; Tokuoka et al., 2011; Xu et al., 2019; Zhang et al., 2016). Given the threat of land abandonment to food security and the rapid development of DPCOS in China, this paper takes China as a case to investigate the impact of DPCOS on land abandonment. We have replaced the title with “Does Disease and Pest Control Outsourcing Service Reduce Arable Land Abandonment? Evidence from China” in the revised manuscript. In addition, although a few studies explore the impact of agricultural outsourcing services on crop production, little is known about what roles do agricultural outsourcing services, particularly DPCOS, play in reducing arable land abandonment.

â‘¢ The purpose of this paper is to investigate the impact of DPCOS on land abandonment. Given the threat of land abandonment to food security, this paper aims at finding an effective way to reduce land abandonment, and thus exploring whether can DPCOS alleviate arable land abandonment.

â‘£We have deleted the conclusions in Introduction Section.

Point 4: The purpose of the work is blurred.

Response 4:  We are sorry for the blurred expression of our purpose. This comment is related to point 3. The purpose of this paper is to investigate the impact of DPCOS on land abandonment. Given the threat of land abandonment to food security, this paper aims at finding an effective way to reduce land abandonment, and thus exploring whether can DPCOS alleviate arable land abandonment. Please find more details in the revised manuscript.

Point 5: The title of part 2 is very laconic; taking into account the content of this part, the question arises what is its role?

Response 5:  Thank you very much for this comment. Disease and pest control outsourcing service (DPCOS) has emerged in the last few years and is likely an effective measure to reduce farmers’ arable land abandonment in China. Thus, Section 2 presents a brief introduction to DPCOS in China. In order to clearly reflect the content of section 2, we have replaced the title with “The development of DPCOS in China”.

Point 6: There is no characterization of the study area (see comment below on ‘Terrain’) in the context of the research, which should be clearly visible in the text.

Response 6Thank you very much for this comment. The samples in our paper are collected from 27 provinces in China except for Hongkong, Macao, Taiwan, Tibet, and Hainan. We use city-level fixed effects and time fixed effects to control both time-varying factors and time invariant factors because the dataset does not have the characteristics of provinces and cities. This paper reported the demographic of household heads (e.g., gender, education level, age as well as hukou and health status) and the characteristics of agricultural production in Table 1.

Point 7: The use of data from 2014 and 2016 raises doubts. What justifies the use of historical data in the undertaken research?

Response 7:  Thank you very much for this comment. â‘ Our research belongs to an empirical analysis that uses historical data of the actual economic system and econometrics tools to test economic theories and explain economic phenomena. Unlike natural science, it is difficult for social scientists to test a conjecture through controlled experiments. Most economic phenomena cannot obtain data through repeated experiments to draw scientific conclusions. Therefore, social scientists have to use historical data to test hypotheses or conjectures. This paper uses the historical data of CLDS to explore the impact of disease and pest control outsourcing services (DPCOS) on land abandonment in China.

 â‘¡China Labor-force Dynamic Survey (CLDS) is a dynamic survey conducted every two years. Although we have learned that CLDS released the data in 2018, we can only get a small part of the information in 2018. The new samples in 2018 cannot be combined with the original follow-up samples in 2014 and S2016 to form a balanced panel data, which will have a greater impact on undertaken research. Therefore, we only use the original follow-up samples in 2014 and 2016 in this paper.

Point 8: Contents of table 1: 'Hukou ?; mu'? Even though there is a reference (Hukou) under the table, it needs to be improved and better related to the contents of the table.

Response 8Thank you very much for this comment. This comment is related to point 1 proposed to be Reviewer 1. First, we have revised the reference to Hukou. The new version is as follows: Hukou refers to the household registration system in China. There is a huge gap between the rural and urban areas in terms of public services, and rural residents can only enjoy urban public services if they get an urban hukou. China established the Hukou system to control the migration of rural residents to urban areas. China has reformed its hukou system since 2014, and a rural-urban unified hukou system has been implemented in some pilot cities. Second, we have added a reference to mu under Table 1. “Mu is a unit of measurement of land area in China, and 15 mu is equal to 1 hectare.”

Point 9: In table 1 –‘Terrain’ - what the Authors mean; how the split was made. How does this relate to the 29 provinces that were available in the database used?

Response 9Thank you very much for this comment. We apologize for not expressing the meaning of ‘Terrain’ in table 1. The ‘Terrain’ is the topographical condition of the villages. CLDS defines terrain as three types, plain, hill, and mountain. The interviewers recorded the terrain type when they visited the surveyed villages. Terrain type is only recorded at the village level and may not be directly related to the 29 provinces. A province usually has all of these three terrain types. In addition, topographical conditions influence outsourcing decisions by changing the cost of agricultural land utilization. For farmers who are not in the village all year round, the more complex the terrain conditions at the village level, the higher the abandonment ratio may be. The development of outsourcing services is also subject to the terrain conditions at the village level. Therefore, it is necessary to consider different topographical conditions in our empirical analysis.

Point 10: There is no discussion of the results.

Response 10: Thank you very much for your suggestions. We apologize for not expressing our research discussion in our manuscript clearly. In fact, we present the discussion and results together in the Section of Empirical Results and Discussion. For example, In general, we may conclude that DPCOS could contribute to the reuse of lands suitable for crop cultivation and the exit of lands unsuitable for crop cultivation. Therefore, DPCOS will also contribute to the formation of multifunctional landscapes discussed by Sturck and Verburg (2016) and Groot et al. (2010).  Please find more details in the revised manuscript.  

Point 11: In the manuscript, I miss an indication of the limitations the Authors see in their research.

Response 11: Thank you very much for your suggestions. We have added the limitations in Conclusion Section. The limitation of this paper is as follows: This study has limitations. There are no household-level data on soil quality and topography in the CLDS dataset, and we have to use village-level data as a proxy. Therefore, all households within a village face the same soil quality and topography.  We would furtherly re-study this topic if we can match CLDS dataset with Harmonized World Soil Database using the longitude and latitude of each household.

Point 12: The work requires intensive improvement of the text editing.

Response 12: We are sorry for the poor writing. We have carefully checked the whole paper and have invited a native speaker to polish our manuscript. Please find more details in the revised manuscript.

Round 2

Reviewer 3 Report

Thank you very much for the authors' corrections and explanations. However, I would like to highlight the following points:

Still, I think that the main purpose of the work is not emphasized enough in the text.

Referring to my remark 7 and the Authors' answers, I suggest that the text of the article be supplemented with a brief explanation of the use of relatively old data, so that the recipient has no doubts as to why the last data for 2016 was used (since we are currently in 2022). What about the data for 2020?

Author Response

Point 1: Thank you very much for the authors' corrections and explanations. However, I would like to highlight the following points: â‘  Still, I think that the main purpose of the work is not emphasized enough in the text. â‘¡Referring to my remark 7 and the Authors' answers, I suggest that the text of the article be supplemented with a brief explanation of the use of relatively old data, so that the recipient has no doubts as to why the last data for 2016 was used (since we are currently in 2022). â‘¢What about the data for 2020?

Response 1: Thank you very much for your comment. â‘ First, we have further emphasized our paper's purpose in the second paragraph of the Introduction section. The revised content is as follows: “Therefore, the purpose of this paper is to investigate the impact of DPCOS on arable land abandonment and to furtherly explore the heterogeneous effects of DPCOS on arable land abandonment across households with different labor endowments and natural resource endowments. Based on the results, this paper will raise some policy implications to reduce arable land abandonment.

â‘¡Second, we have added a brief explanation of the use of relatively old data as a footnote in the Methodology and Data section. The added footnote is as follows: “The CLDS data for 2020 has not been released yet. Although we have learned that CLDS released the data in 2018, we can only get a small part of the information in 2018. The new samples in 2018 cannot be combined with the original follow-up samples in 2014 and 2016 to form balanced panel data, which will have a greater impact on undertaken research. Therefore, this paper only uses the original follow-up samples in 2014 and 2016.

â‘¢ The CLDS data for 2020 has not been released yet.
